# Cryptocurrencies and Fraudulent Transactions: Risks, Practices, and Legislation for Their Prevention in Europe and Spain

**David Sanz-Bas** [1,*], **Carlos del Rosal** [2], **Sergio Luis Náñez Alonso** [3] and **Miguel Ángel Echarte Fernández** [3,*]

1 Department of Economics, Catholic University of Ávila, Canteros St., 05005 Ávila, Spain
2 Independent Researcher, 05005 Ávila, Spain; carlosdelrosal2021@gmail.com
3 DEKIS Research Group, Department of Economics, Catholic University of Ávila, Canteros St., 05005 Ávila, Spain; sergio.nanez@ucavila.es
* Correspondence: david.sanz@ucavila.es (D.S.-B.); mangel.echarte@ucavila.es (M.Á.E.F.)

**Abstract:** Cryptocurrencies have been developing very rapidly in recent years, and their use is becoming more and more widespread in different areas. The use of digital currencies for legal uses is advancing along with technological development, but, at the same time, criminal activities are also emerging to take advantage of this boom. The aim of this paper has been, first, to analyze the various ways in which individuals and criminal organizations have taken advantage of the phenomenon of cryptocurrencies to carry out fraudulent activities such as laundering money of illicit origin and, second, to provide an overview of the legal tools that have been developed in this regard in Europe and, more specifically, in Spain to combat these activities. Undoubtedly, cryptocurrencies bring great benefits to the economy, but it is also necessary to know the risks and abuses that have been developed to prevent them.

**Keywords:** cryptocurrencies; crypto-crime; crypto-legislation; risks; fraud practices; prevention; risk reduction





## 1. Introduction

In recent years, there has been a very rapid development of digital currencies or cryptocurrencies, with different types existing such as Bitcoin, Ethereum, Litecoin, Dogecoin, Ripple, etc. (Bouri et al. 2019). Currently, different uses of these cryptocurrencies are common, since they can be used as an investment, deposit, to make transfers and to make payments. A cryptocurrency is a decentralized digital asset that uses cryptography to secure transactions between users. One of its advantages is that its value is independent of the monetary policies of central banks and that it is based on blockchain technology (Sanz Bas 2020; Echarte Fernández et al. 2021). Governments, for reasons of security, the control of monetary policy, cash alternatives, etc., are considering the creation of state digital currencies (Náñez Alonso et al. 2020).

Despite the security offered by this type of digital currencies, we must realize that on many occasions they are a tool used to commit fraud and are being used for tax evasion (often due to the ignorance of tax regulations), carrying out illegal transactions or money laundering (Sanz Bas 2020; Náñez Alonso 2019; Cheah and Fry 2015; Dyntu and Dykyi 2019; Nikolova 2019; Kfir 2020; Wronka 2021).

While there is doubt about its use for fraudulent transactions, there is currently a debate around it (Dyson et al. 2018; Butler 2019). In this sense, a study was carried out on a dataset of 4681 accounts of the Ehtereum network and it turned out that 2179 were illicit (Farrugia et al. 2020). Another study detected 274 cases of fraud within the 1393 Initial Coin Offerings (ICOs) studied (Hornuf et al. 2021). Other studies point out that Bitcoin is low risk for money laundering (Navarro Cardoso 2019), or also indicate that out of a market of USD 500 billion, only USD 10 billion is used in scams and other illegal activities (2% of the total) (Geography of Cryptocurrency Report 2020).

One case to highlight is the one concerning the Distributed Autonomous Organization (DAO), which was created on Ethereum in the spring of 2016. The DAO experiment failed shortly after its creation, as an anonymous hacker stole more than USD 50 million in Ethers out of the USD 168 million invested (Shier et al. 2019).

Faced with this situation and in the interest of ensuring user protection of these new means of payment, central banks have reacted. To this end, they are considering the issuance and implementation of a Central Bank Digital Currency (CBDC), as the Bahamas has done (Náñez Alonso et al. 2021). It is also worth noting the possible implications and the great challenge that the emerging system of decentralized finance poses for the current financial system (Echarte Fernández et al. 2021).

The practice of hiding income from illicit activities dates back to the Middle Ages, with pirates being pioneers in the sixteenth and eighteenth centuries, through the commercial ships that sailed the Atlantic. Part of the treasures and wealth accumulated by the pirates was kept in lairs, thus giving rise to financial safe havens (Tondini 2009). In the traditional banking system, there is an intermediary, the bank, and supervision by state authorities; in cryptocurrencies, these actors disappear. There is no central authority to regulate it, functioning as a community, with the users themselves validating the decisions through a system of blocks, called blockchain, which are formed by deciphering mathematical algorithms (Porxas Roig and Conejero 2018).

Money laundering constitutes one of the problems of our society, due to its economic, political, and social consequences (Martínez 2017; Albrecht et al. 2019; Chowdhury 2019). Technological development is fundamental in the emergence of cryptocurrencies, surfacing to avoid dependence on traditional financial systems. According to Law 10/2010, of 28 April 2010, on the prevention of money laundering and the financing of terrorism[1], money laundering can be defined as "the set of mechanisms or procedures aimed at giving the appearance of legitimacy or legality to goods or assets of criminal origin". With cryptocurrencies, tools and services have been developed that can be used for illegal purposes. The anonymity or pseudo-anonymity (depending on the Blockchain used) in the ownership of the cryptocurrency, using different methods such as "mixers" and "exchanges", hinder, in some cases, the investigation and traceability of operations, becoming a facilitating agent for money laundering. The main characteristics of cryptocurrencies are security, speed, based on cryptography, pseudo-anonymous, decentralized, the elimination of intermediaries, and the facilitation of international transactions (Barroilhet Díez 2019). All these characteristics make cryptocurrencies an attractive and useful product for money laundering.

In our article, however, we do not try to prove whether a lot or little fraud is committed through the use of cryptocurrencies, nor do we try to stigmatize their use, purchase, and sale. We approach this question from a double point of view: first, we analyze which are the most common cases of fraud and money laundering through the use of cryptocurrencies, and second, what response is currently given by Spanish regulations (both criminal and related to the prevention of fraud). For this purpose, our article is structured as follows: First, an approach to cryptocurrencies is outlined. Second, we address the legislation applicable to cryptocurrencies in Spain and in Europe, both domestic and from international organizations. Third, we analyze the figure of money laundering; ending with an analysis of the methodologies used to carry out money laundering with cryptocurrencies.

## 2. Approach to Cryptocurrencies

"Cryptocurrencies are offspring of the digital revolution that has taken place over the last few decades" (Sanz Bas 2020, p. 15). Cryptocurrencies are digital or virtual currencies that use cryptography for their security and are not issued by any central authority (López Domínguez and Melón 2020). "They could be defined as a type of digital currency that

---

[1]　Authors's translation. In Spanish: "Ley 10/2010, de 28 de abril, de prevención del blanqueo de capitales y de la financiación del terrorismo. «BOE» núm. 103, de 29/04/2010".

is based on cryptography and on the Blockchain technology" (Sanz Bas 2020, p. 17). Nakamoto (2008) defines an electronic coin "as a chain of digital signa-tures. Each owner transfers the coin to the next by digitally signing a hash of the previous transaction and the public key of the next owner and adding these to the end of the coin". In 1998, the first electronic currency emerged, when Wei Dai, a renowned cryptographer and member of the cypherpunk community, who is dedicated to digital activism while protecting his privacy and security, spread a mailing list called "cypherpunk" for the exchange of securities. This exchange is based on an electronic currency that could not be traced and whose users remained anonymous, this currency was called "b-money" and shows great similarities with the current cryptocurrencies (López Domínguez and Melón 2020; Ordinas 2017). Already in 2009, and as a consequence of the liquidity crisis in the financial markets that same year, Satoshi Nakamoto, the pseudonym of a user or group of users, published "The White Paper" entitled "Bitcoin: a user-to-user electronic cash system"; in this do-document the foundations of the Bitcoin system are laid (Ordinas 2017). A new type of money is born, which, as we have already mentioned, uses cryptography to allow user-to-user economic transactions without verification by a third party, without centralized authority, and with a very high security protocol (Mora 2016). The popularity of Bitcoin grows very fast and, subsequently, numerous digital coins are created, which have different features and protocols, but are essentially similar and based on the original currency, Bitcoin (Mora 2016). It is currently a booming market, with an increasing number of virtual coins with a high market value. This development of cryptocurrencies is associated with a debate on whether or not central banks should issue digital money (Náñez Alonso 2019). Currently, the main cryptocurrencies are private and highly volatile, which makes it difficult for them to be used as a general medium of exchange (Náñez Alonso et al. 2020). To understand the world of cryptocurrencies, it is necessary to define a series of basic concepts such as virtual currencies, cryptocurrencies/cryptoassets, blockchain, wallet, or bitcoin.

The Financial Action Task Force (FATF 2015, p. 28) defines virtual currencies as "a digital representation of value that can be traded digitally and functions as a medium of exchange, a unit of account and a store of value, but does not have legal tender status in any jurisdiction. It is neither issued nor guaranteed by any jurisdiction and fulfills the above functions only by agreement within the community of virtual currency users". This definition can be completed with the one provided by the European Central Bank [(ECB)] (ECB 2012, p. 5), which states that virtual currencies are "a type of unregulated, digital money, which is issued and generally controlled by its developers and used and accepted among the members of a specific virtual community". Therefore, virtual money or currency can be converted into physical coins or bills but does not have legal tender status in most countries. Recently, the Legislative Assembly of El Salvador has given Bitcoin the status of legal tender, but thus far it is the only country that has done so (CoinMarketCap 2021b). Additionally, there is a community behind it that supports it, and there are developers who create and control it.

Cryptocurrency or cryptoasset is understood as a "decentralized, distributed, con-vertible virtual currency, mathematically based on a complex algorithm and that uses a cryptographic system to ensure the integrity of the transactions of the system on which it is based. As a general rule, there is no control system, central bank, or centralized storage, since it uses decentralized, shared, and synchronized networks" (Hancock and Vaizey 2016). The functioning of cryptocurrencies is based on a system of public and private keys that serve to carry out transactions reliably and securely, and each transaction must be signed cryptographically (Marrero Travieso 2003). Security is guaranteed and most of them make use of blockchain technology. Cryptocurrencies allow users to transfer value directly from one to another without having to resort to an external intermediary, thus bypassing the banking system (Palomo-Zurdo 2018). Blockchain is the basis or support for cryptocurrencies, and especially for Bitcoin; its concept refers to a database in which each user, with each transaction he/she executes, issues information that is aggregated in data blocks (Corredor Higuera and Guzmán 2018; Parrondo 2018). Unlike the banking

system, in which there are intermediaries, with blockchain, which replaces traditional financial institutions, we are faced with a secure, transparent, and decentralized system, in which there are no intermediaries or supervisors (Parrondo 2018). The blockchain is a system that verifies and records the operations of a cryptocurrency, decentralizing the entire management. One could say that it is a digital ledger, which ensures the integrity of its contents (Argañaraz et al. 2019). The blockchain operation scheme is shown in Figure 1.

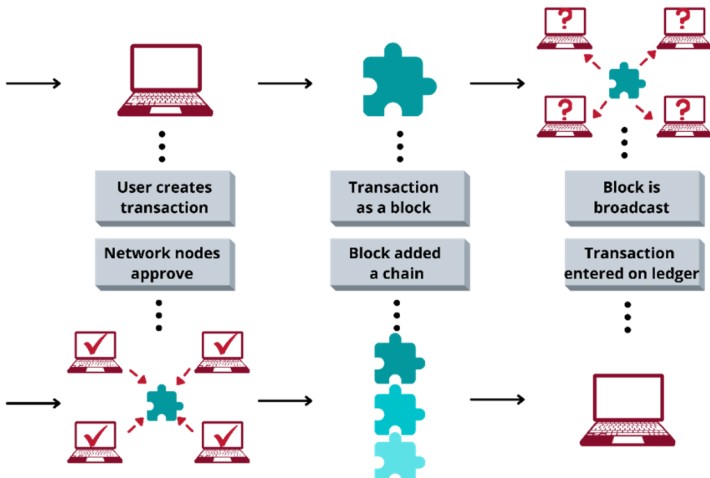

**Figure 1.** Blockchain operation. Source: Own elaboration.

The operations carried out with blockchain technology, in principle, would allow knowing the transactions that a person makes, since they are permanently recorded; however, there are ways to hide the operations (Barroilhet Díez 2019). Block records are linked and encrypted, thus protecting the security and privacy of transactions, with users verifying transactions. Blockchain organizes cryptocurrencies and acts as a ledger where transactions are recorded (Enguix 2020). The term Blockchain should not be confused with Wallet, which is a virtual wallet or purse. It is a software, offered by several internet companies, that manages and stores the public and private keys of cryptocurrencies, which allows access to a user's cryptocurrency by entering keys through an accretion system (Gallardo et al. 2019). There are wallets that allow access to a single cryptocurrency or to several. Bitcoin is the first and most popular digital currency, as can be seen in Figure 2.

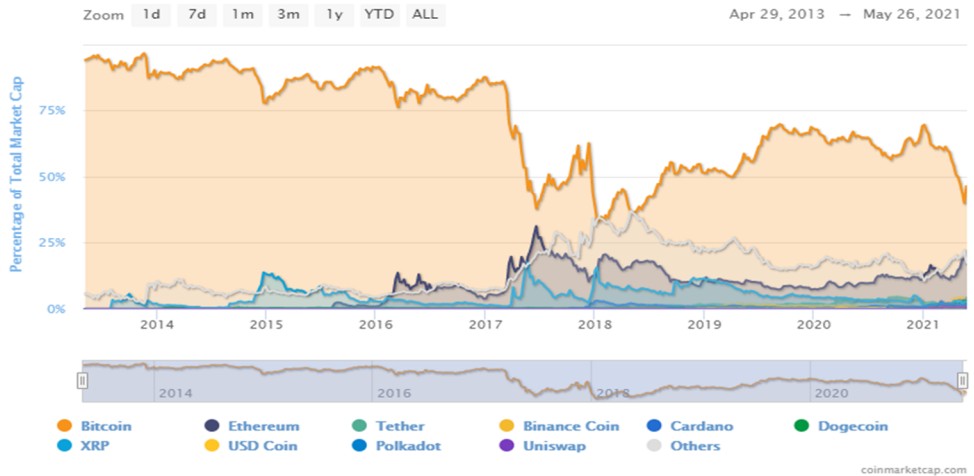

**Figure 2.** Market quota of the main cryptocurrencies by market capitalization. Source: CoinMarket-Cap (2021a).

Bitcoin is characterized by being a pseudo-anonymous currency and having an open and free access system, anyone can join without any authorization. The accounts are kept by the community itself, being a system based on trust (Barroilhet Díez 2019). Its acceptance is quite widespread as a form of payment and, in numerous exchanges, it can be exchanged for legal tender (Bedecarratz Scholz 2018). A shift in investment is occurring globally, it is observed in that many users are using Bitcoin as an investment, the process of replacing gold with Bitcoin in the portfolios of many institutions is accelerating, and Bitcoin in 2021 is becoming a global digital store of value (Bloomberg Galaxy Crypto Index [BGCI] 2021). Since Bitcoin's price is highly volatile (cf. Figure 3), Bitcoin's market cap suffers also significant fluctuations.

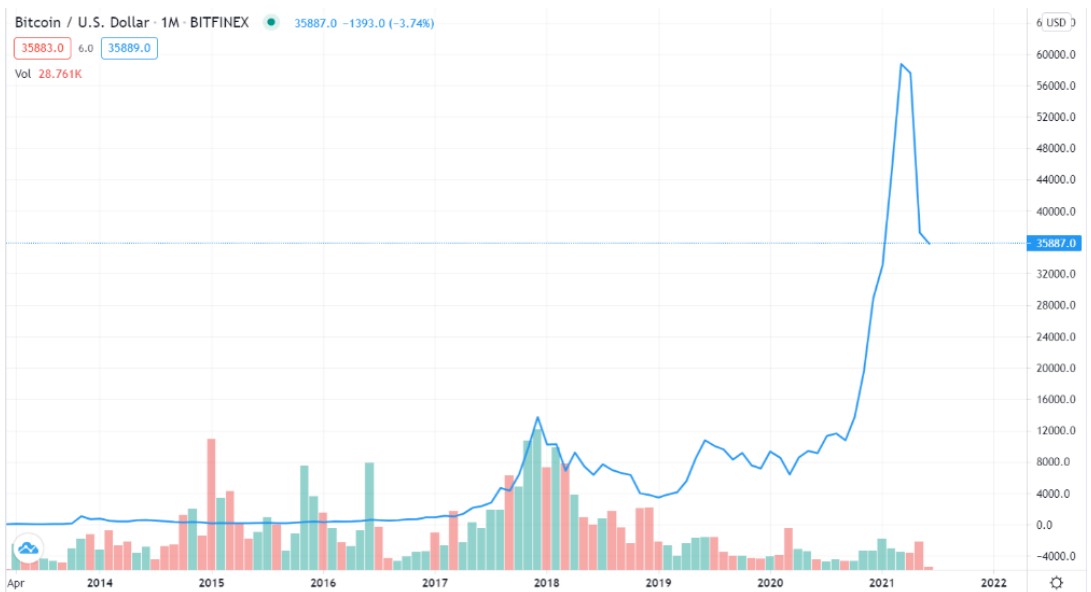

**Figure 3.** Bitcoin price evolution. Source: Tradingview (2021).

## 3. Legislation Applicable to Cryptocurrencies in Europe and Spain

As we have analyzed in the introduction, the uses of cryptocurrencies are not always legitimate, and despite the apparent security offered by virtual currencies, we find the existence of their fraudulent use. Legislation has had to adapt (Teichmann and Falker 2020; Podgor 2019).

As regards the European Union, on 30 May 2018, Directive (EU) 2018/843 (EU 2018), also known as the "Fifth Directive", on the prevention of the use of the financial system for the purpose of money laundering, was published, establishing concepts related to virtual currencies and guiding countries on how national regulations should be. This EU directive determined for its transposition in the member countries, the date of 10 January 2020; however, in Spain this transposition was published on 27 April 2021. In addition to the aforementioned legislation, FATF published in 2019 a Guide on Virtual Assets and Service Providers. This organization often makes recommendations, publications, reports, and evaluations to certain countries that lead to European regulations, which, in turn, materialize in national legislation. In this sense, the information on FATF publications and reports allows us to know which route the Spanish financial system is going to follow. In this sense, the following would be the reference publications at an international, European, and national level, respectively, in the regulation of virtual currency:

— FATF Guide on Virtual Assets and Service Providers, published in June 2019.
— Fifth Directive, 2018/843 EU, on the Prevention of Money Laundering.

— Royal Decree-Law 7/2021 transposing EU directives, including the one on the prevention of money laundering (Royal Decree-Law 7/2021 Transposing EU Directives, Including the One on the Prevention of Money Laundering 2021)[2].

The guide on Virtual Assets and Service Providers (FATF 2019) provides us with two important definitions such as Virtual Asset (VA) and Virtual Asset Service Provider (VASP). The former is the digital representation of value that can be traded or transferred digitally and can be used for payment or investment purposes. Additionally, VASP is that natural or legal person that is not covered elsewhere under the recommendations and performs any of the following activities: inter-exchange between virtual assets and fiat currencies, exchange between one or more forms of virtual assets, the transfer of virtual assets or participation, and the provision of financial services related to the offer and/or sale of a virtual asset. In addition, the following different fundamental points are set forth in this guide and catalogued as an interpretative note:

— VA's consideration is property, income, funds. It is a risk-based approach, risks must be identified, and preventive and repressive measures must be taken.
— VASPs should be required to be licensed or registered.
— At a minimum, VASPs should be licensed or registered in the jurisdictions where they are created.
— When the VASP is an individual, they should be required to be licensed or registered in the jurisdiction where their place of business is located.
— Jurisdictions may require VASPs offering products in their jurisdiction to be registered in that jurisdiction.
— VASPs must be subject to appropriate regulation and supervision.
— ASPs must be controlled by a competent authority that should conduct risk-based supervision, have adequate powers to supervise, conduct inspections, compel the production of information, and impose sanctions.

Penalties, whether criminal, civil, or administrative, should be available for dealing with non-compliant VASPs. The preventive measures to be put in place are as follows:

○ VASP: CDD (Customer Due Diligence) from 1000 USD/EUR.
○ VA: Accurate information, payer-beneficiary, and make it available at the request of the authorities.
○ Measures: Blocking, prohibition of transactions with persons and entities.
○ International cooperation, VASP supervisors should exchange information quickly and constructively with their foreign counterparts.

The fifth directive is the 2018/843 of the European Parliament and of the Council dated 30 May 2018 (EU 2018) on the prevention of the use of the financial system for the purpose of money laundering or terrorist financing, and amending Directives 2009/138 EC and 2013/36UE, reflects several aspects on cryptocurrencies, which have been transposed into our legislation and we consider that the most noteworthy would be the following:

— It incorporates new obligated parties: the providers of services for the exchange of virtual currencies for fiat currencies (cryptocurrency exchange operators or exchanges) and the providers of custody services for electronic wallets or private keys or wallets or electronic wallets (entities that provide services for the safeguarding of private cryptographic keys on behalf of their clients, for the holding, storage, and transfer of virtual currencies).
— It aims to make providers report suspicious transactions, in addition to partially restricting the anonymity that cryptocurrencies supposedly allow. The competent

---

2  Authors's translation. In Spanish: "Real Decreto-ley 7/2021, de 27 de abril, de transposición de directivas de la Unión Europea en las materias de competencia, prevención del blanqueo de capitales, entidades de crédito, telecomunicaciones, medidas tributarias, prevención y reparación de daños medioambientales, desplazamiento de trabajadores en la prestación de servicios transnacionales y defensa de los consumidores. 2021. «BOE» núm. 101, de 28 de abril de 2021, páginas 49749 a 49924".

authorities should be empowered, through the obliged entities, to monitor the use of virtual currencies.

— Another measure in relation to these new subjects is the obligation for them to be registered, although it is not specified in what type of registration or the terms and conditions of this. The Directive itself sets a date of January 10, 2020 for its transposition.

As regards the applicable Spanish regulations, we must start from Law 10/2010 of April 28 on the Prevention of Money Laundering and Terrorist Financing (2010), as well as the following regulations that updated it, which did not regulate the activity of cryptocurrencies. As stated by the Bank of Spain (BE) and the National Securities Market Co-mission (CNMV) (BE and CNMV 2021, p. 2), in February 2021, in the European Union there is still no "framework regulating cryptoassets such as Bitcoin and providing guarantees and protection similar to those applicable to financial products. Currently, a Regulation (known as MiCA) is being negotiated at the European level that aims to establish a regulatory framework for the issuance of cryptoassets and service providers on these." They further state that, from a legal point of view, cryptocurrencies cannot be considered as a means of payment, are not backed by authorities or a central bank, and are not covered by mechanisms that protect customers such as the Investor Guarantee Fund or the Deposit Guarantee Fund (BE and CNMV 2021).

On 12 June 2020, the preliminary draft amendment to the Law 10/2010 of April 28 on the Prevention of Money Laundering and Terrorist Financing (2010) was published on the website of the Ministry of Economic Affairs and Digital Transformation. Additionally, on 27 April 2021, the Royal Decree-Law 7/2021 on the transposition of European Union directives, including the one on the prevention of money laundering, was published. Therefore, Directive 2018/843 of the European Parliament and of the Council (EU 2018), also called the fifth directive, is transposed.

On 27 April 2021, Royal Decree-Law 7/2021 was published, transposing European Union directives, including the one on the prevention of money laundering, which establishes the regulatory framework for virtual currencies, setting specific deadlines for its entry into force. It is noted that the activity of intermediation in the sale and purchase of cryptocurrencies entails a great risk of money laundering, which, until now, had not been regulated. This Royal Decree-Law 7/2021 establishes the regulatory framework for cryptocurrencies, regulates the obligated parties, the supervisory and regulatory entity, and presents different definitions, among them the virtual currency and electronic wallet custody service providers. It also regulates the Registry of service providers for the exchange of virtual currency for fiat currency and establishes a sanctioning regime by the Bank of Spain. The most relevant points of this transposition regarding virtual currencies are as follows:

— New obligated parties are incorporated, including electronic wallet custody service providers, which are defined as "those natural persons or entities that provide safeguarding or custody services of private cryptographic keys on behalf of their clients for the holding, storage and transfer of virtual currencies".
— Persons providing services for the exchange of virtual currency into legal tender must be subject to the preventive obligations established.
— Virtual currency is defined as "a digital representation of value not issued or guaranteed by a central bank or public authority, not necessarily associated with a legally established currency and which does not have the legal status of currency, but which is accepted as a medium of exchange and can be transferred, stored or traded electronically". The exchange of virtual currency for fiat currency is understood as "the purchase and sale of virtual currencies through the delivery or receipt of euros or any other foreign currency of legal tender or electronic money accepted as a means of payment in the country in which it was issued".
— The Registry of service providers for the exchange of virtual currency for fiat currency and the custody of electronic purses is regulated. Natural or legal persons, regardless

of their nationality, who offer or provide in Spain services for the following must be registered in the registry set up at the Bank of Spain:

— Sale of virtual currencies through the delivery or receipt of euros or any other foreign currency of legal tender or electronic money accepted as a means of payment in the country in which it has been issued.
— Providers of electronic purse custody services.
— The following must be registered: "Natural persons who provide these services, when the base, direction or management of these activities is located in Spain, regardless of the location of the recipients of the service" and "Legal persons established in Spain who provide these services, regardless of the location of the recipients".
— Supervisory powers are delegated to the Bank of Spain, related to compliance with the obligation to register and the conditions of honorability of registration.
— A sanctioning regime is established by the Bank of Spain, when the provision of services is carried out without the mandatory registration as a very serious infringement, being serious if the activity has been carried out on an occasional basis.
— The entry of this law into force is established on the day following its publication in the BOE; however, several exceptions are established, related to virtual currencies.

In the second transitory provision, a term of six months is established from the entry into force of this royal decree-law, for the registration in the registry of providers of services of exchange of virtual currency for fiat currency and custody of electronic wallets.

This transitional transposition also establishes a maximum period of nine months for the registration in the register of the Bank of Spain of individuals or legal entities that are providing any of the following services to residents in Spain:

— Sale of virtual currencies through the delivery or receipt of euros or any other foreign currency of legal tender or electronic money accepted as a means of payment in the country in which it has been issued.
— Electronic wallet custody service providers.

Both the FATF Guide, the Fifth Directive and the Royal Decree-Law are aimed at defining virtual assets and establishing virtual asset service providers as regulated entities. The three publications are based on a risk-based approach, in which the obliged parties must carry out due diligence measures on customers, establishing a preventive and a repressive or sanctioning system, with international cooperation being essential for a rapid and effective exchange of information.

Regarding the European regulation of crypto-assets, cryptocurrencies, and Tokens in September 2020 the European Commission presented the communication COM (European Commission 2020) 593: Proposal for a REGULATION OF THE EUROPEAN PARLIAMENT AND OF THE COUNCIL on Markets in Crypto-assets, and amending Directive (EU) 2019/1937 (known as MiCA). This is a pioneering regulatory proposal, which will affect the following types of crypto-assets:

1.  Utility tokens: these are those that are normally used to obtain initial funding for the development of a specific project, used, for example, by some startups. They can be issued without the need for authorization, being sufficient to notify the competent national authority (in Spain, probably the CNMV) and submit a White Paper (the document containing all the technical details and description of the product).
2.  Asset-Referenced Token: These cryptoassets serve as a medium of exchange, as they aim to keep stable the value of one or several commodities or one or several cryptocurrencies, or a combination of both, referenced to several national currencies of legal tender. In order to authorize their issuance, it is necessary to approve their White Paper and comply with certain requirements.
3.  Electronic money token (e-money token): This is another cryptoasset used as a medium of exchange, but whose value is intended to be maintained by denominating it in units of a national currency. For its issuance, a notification must be sent to the competent authority and be a credit or e-money institution.

The future MiCA regulation will also affect cryptoasset service providers. MiCA establishes the obligation for them to be authorized by a competent authority to provide this type of service. Such authorization will be subject to the fulfillment of a series of requirements, such as having a certain capital fund, having, as we have already said, a registered office in the EU, having insurance, qualified management personnel, a custody system for client funds, and even a business continuity plan. Apart from these requirements, additional requirements will also be established for cryptoasset service providers that

- Have custody of and provide access to cryptoassets to their clients.
- Have cryptoasset exchange platforms.
- Execute cryptoasset purchase orders.
- Issue cryptoassets (when they are both the issuer and service provider).
- Receive and transmit orders related to cryptoassets.
- Provide advisory services in the field of cryptoassets.
- Make payments of tokens referenced to assets.

However, as of today, the regulations have not yet been approved. It is expected to be approved by the end of 2021 or early 2022.

## 4. Money Laundering

Before analyzing the meaning of money laundering, its regulatory provisions, and its characteristics, we must point out that money laundering is a means of maintenance and survival of criminal organizations, which generates a lot of cash from different illegal activities that they have to introduce into the legal circuit to be able to use it and benefit from it. Therefore, the fight against money laundering has a broader scope than the investigation of specific types of crimes. Money laundering has to be linked to a criminal activity (drug trafficking, arms trafficking, human trafficking, terrorism, tax fraud, corruption, etc.), since on its own in Spain it is not criminal (Fernández Bermejo 2016). The dismantling of a criminal organization, materialized only in the arrest of individuals who have participated in criminal offenses, is not the most complete and effective response, since organizations restructure quickly. Analyzing, blocking, and intervening in a criminal organization's financial system (corporate networks, trusts, bank accounts, movable, and immovable assets, etc.) can weaken and, in some cases, paralyze its illicit activity (FATF 2012).

For the FATF (2018), money laundering can be defined as the processing of criminal proceeds to disguise their illegal origin. The regulations governing money laundering can be divided in two; on the one hand, there is the preventive one, focused on the non-commission of infractions and on the other hand the repressive or punitive one, which is when the acts carried out lead to an infraction regardless of the stage in which the illicit is found.

Under the Spanish law, money laundering is included in Organic Law 10/1995, of 23 November 1995, of the Penal Code, Title XIII, Chapter XIV[3], regulated in Articles 301 to 304 of the Penal Code in force. Article 301.1 contains the following typical description "whoever acquires, possesses, uses, converts or transmits assets knowing that these have their origin in a criminal activity, committed by him or by any third person or performs any other act to conceal or cover up their illicit origin" (Organic Law 10/1995, of 23 November 1995, of the Penal Code 1995). In money laundering activities, legal entities are used as instrumental entities for the commission of money laundering crimes, sometimes being created for that exclusive purpose, or else, to mix illicit funds with those coming from a real activity. There are law firms or law offices dedicated to advising on tax fraud and money laundering, which are in charge of designing the companies and creating bank accounts in the countries required, according to the characteristics of the client.

Therefore, the conducts typified in article 301 of the Organic Law 10/1995, of 23 November 1995, of the Penal Code (1995) are constitutive of money laundering if they are aimed at hiding or concealing their illicit origin or to help the person who has participated

---

[3] Authors's translation. In Spanish: "Ley Orgánica 10/1995, de 23 de noviembre, del Código Penal. «BOE» núm. 281, de 24/11/1995".

in the infraction or infractions to evade the legal consequences of their acts. To understand the crime under analysis, it is necessary to know that it is a crime of mere activity (it does not need a result to be consummated), with different phases from the time the criminal organization has the money in cash until it is laundered and put into circulation. The doctrine accepts the theoretical division of the operation of money laundering in three phases, agreeing also that many times such division may appear unrecognizable.

The phases overlap, are simultaneous, and disorderly, with the methods of laundering becoming increasingly complex, and some of the phases may be repeated. The doctrine, following a terminology used by the FATF (2018), on Money Laundering agrees that this crime generally develops in the following three stages: First, placement; second, layering or interleaving or diversification or conversion; and finally, integration or investment. The three stages are briefly described in the following below:

1.  Placement: This is the first stage of the laundering process, in which the "dirty" money, whether in cash or materialized in any type of illicitly obtained goods, changes location, and is placed beyond the control of the authorities.

2.  Layering or intermingling or diversification or conversion: This is the second stage of the money laundering process and consists of intermingling the money or goods in various businesses and financial institutions. The money is transported to other places to disguise its illicit origin. The important thing here is to acquire assets in order to transfer or exchange them with others of licit origin. At this stage, once the money enters the financial circuits, movements are made to hide its origin, in order to eradicate any possible link between the money placed and its origin. The most frequent techniques are to send the money to tax havens or offshore centers, to ensure that the funds circulate throughout different countries, institutions, and accounts held by different individuals or legal entities.

3.  Integration or investment: This is the last stage of the money laundering process. The money from criminal activities is used for financial business, which may be legitimate. In this stage, commercial investments are made, loans are granted to individuals, goods are purchased, and all kinds of transactions are carried out through accounting and tax records, which justify the capital in a legal manner, making control difficult. Therefore, the money is placed back into the economy, with the appearance of legality. In this phase, once the capital has been placed and stratified, the money returns to the legal financial circuit mixed with other licit elements, thus giving it the appearance of legality.

Figure 4 below shows the three stages mentioned above.

FATF (2018) provides a wealth of information on various illicit activities, including money laundering and cryptocurrencies, conducting prospective analyses that make it possible to anticipate and forecast the future in the matters covered. The FATF develops recommendations to prevent organized crime, corruption, and terrorism, ensuring a worldwide response in a coordinated manner and monitoring countries for their actual implementation.

Its objectives are the development of a common model for the fight against money laundering (40 recommendations), the control and improvement of national prevention systems through periodic evaluations of countries, the review of new money laundering techniques, and the international cooperation network through regional bodies (GAFILAT, CFATF, ASIA-PACIFICO, EUROPE) and the identification of non-cooperative jurisdictions (NCTCs).

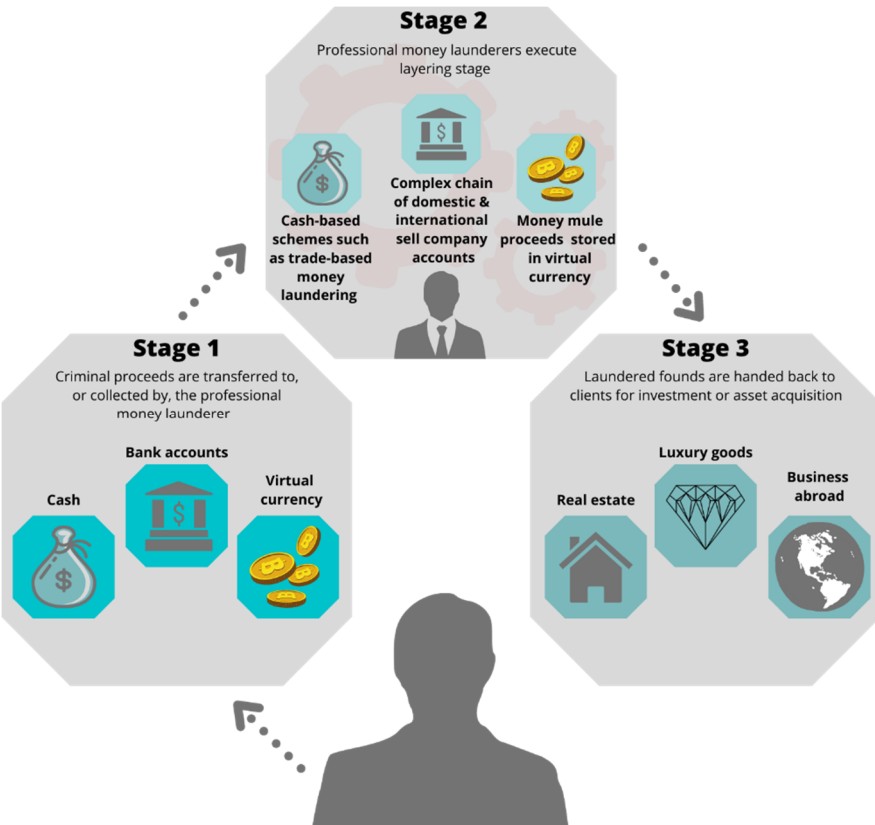

**Figure 4.** Three stages of professional money laundering. Source: Own elaboration based on (FATF 2018).

## 5. Methodologies for Money Laundering with Cryptocurrencies

The FATF reviews money laundering and terrorist financing techniques, continually strengthening its standards to address new risks, such as the regulation of virtual assets, carrying out prospective analyses that make it possible to anticipate and forecast the future in the matters dealt with. The old types of laundering used by organizations, such as the use of cash, have been evolving. Currently, there is a preventive and repressive system of money laundering, which prevents certain operations from being carried out or, if they are carried out, certain institutions are warned by means of automatic reports. Limits are established to buy products and services through cash, the banking system has set limits when carrying out different transactions, which greatly limit the possible payment in cash. Faced with this situation, criminal organizations are updated, either they have specialized personnel, or they contact organizations dedicated to money laundering. Through cryptocurrencies, new methodologies for money laundering are emerging, making it difficult for countries to fight against this type of organized crime. The risks arising from the use of cryptocurrencies for illicit purposes can be grouped into the following three sections:

1.  Use of cryptocurrencies to pay for criminal services: a whole series of crimes linked to cybercrime are encompassed, in which payments are made through these currencies. Illegal transactions take place on the dark internet (Darknet), including the purchase and sale of narcotic substances, counterfeit banknotes and products and false identity documents, the sale of cloned or stolen bank card data, software containing viruses (malware), images and videos with pedophile content, weapons, and explosives. For example, Silken Road, an illegal market within the Darknet, in which the sale and purchase of illicit products was carried out through cryptocurrencies, was closed by the Federal Bureau of Investigation (FBI) on 2 October 2013.

2.  Use of virtual currencies as a form of fraud: The revaluation that virtual currencies have had in recent years, has led to the emergence of new forms of business through ICOs (Initial Coin Offer). "An initial coin offering (ICO) is the cryptocurrency industry's equivalent to an initial public offering (IPO). A company looking to raise money to create a new coin, app, or service launches an ICO as a way to raise funds...Initial Coin Offerings (ICOs) are a popular fundraising method used primarily by startups wishing to offer products and services, usually related to the cryptocurrency and blockchain space". (...) "No 'cryptocurrency' issuance nor any ICO has been registered, authorized or verified by any supervisory body in Spain. This implies that there are no 'cryptocurrencies', or 'tokens' issued in ICOs whose acquisition or holding in Spain can benefit from any of the guarantees or protections provided for in the regulations relating to banking or investment products" (BE and CNMV 2018, p. 2; cf. Frankenfield 2020). Among the priorities of the BE and the CNMV is to "provide information to the public so that investors and users of financial services are in a position to face the increasing complexity of the financial environment with confidence" (BE and CNMV 2018, p. 2). They consider that it is essential that all individuals or entities that decide to buy digital currencies or invest in these products, consider the risks associated with them, and that the decision is made with all the available and sufficient information to understand the product they are going to acquire.

3.  Use of virtual currencies for laundering money of illicit origin: First, the money generated in an illicit activity is transformed into virtual currency. Second, operations are carried out to erase the traceability of the cryptocurrency, such as the use of mixers. Third, after the stratification phase, it is introduced into the real circuit to monetize these illicit profits. The criminal organization that generates the illegal money from some kind of criminal activity may have the structure and know-how to perform the self-laundering of that money or contact an organization that is dedicated to laundering the money, co-charging a commission in proportion to the amount laundered. According to reports from the following different agencies analyzed by Navarro Cardoso (2019): National Crime Agency (NCA), United Nations Office on Drugs and Crime (UNODC), and United States Drug Enforcement Agency (DEA), it is found that, due to anonymity, speed, transnationality, and non-presence, new technologies have favored criminal activity in the field of money laundering and tax fraud, in addition to other types of activities (means of payment in drug trafficking or extortion). It is clear from these reports that virtual currencies are becoming a relatively safe method for criminals to move illicit profits around the world with a lower risk than traditional methods (Navarro Cardoso 2019). The different ways of laundering money through cryptocurrencies are presented below.

*5.1. Mining*

Mining in the world of cryptocurrencies can be defined as the set of processes that are necessary to process and validate the transactions of a cryptoasset using a blockchain network. The protocol only allows these transactions to be processed by specialized users called miners (Barroilhet Díez 2019). Within such a network, for a transaction to be validated, a complex mathematical problem must be re-solved. The key must be decrypted by performing random in-tents, gaining the right to decide the block; noting that block in the ledger, these computers are the miners of the cryptocurrency.

Once the mathematical problem has been deciphered, the transaction is added as another block, becoming irreversible. The first user who manages to solve the mathematical problem receives a reward in the form of a number of cryptocurrencies, at which point these coins enter into circulation. The miners, apart from unlocking coins and adding them to the network, also check the operations performed. Virtually all cryptocurrencies are generated through mining, whereby a number of users, the miners, are in charge of performing the aforementioned tasks.

Mining equipment is quite expensive, more than 1500 euros per unit, very noisy, and consumes a great deal of energy. Currently, it is necessary to have a great technological capacity to be able to mine virtual currencies; therefore, it is necessary to have a great computing capacity to mine cryptocurrencies simultaneously. It is necessary to create a large infrastructure to install them in a soundproof place.

For these reasons, companies or organizations that carry out this activity tend to choose countries where electricity is cheaper or colder countries where less cooling is needed. According to Jiang et al. (2021), 75% of BTC mining occurs in China due to the proximity to hardware manufacturers and lower electricity costs.

Organizations that obtain illicit money through various criminal activities and need to launder it, use such money to purchase mining equipment, in order to mine cryptocurrencies. In addition, since large electricity consumption is required, mining can often lead to a related crime of electricity fraud.

Once the profits are obtained through mining, they could be reinvested in any asset or in an apparently legal activity, the benefits of which can be justified to the authorities.

### 5.2. Exchanges

These are virtual platforms that facilitate the exchange of cryptocurrencies to ensure that customers can transform real money into virtual money and vice versa, through various payment mechanisms, such as bank transfers, credit cards, PayPal, etc.

Exchanges act as wallets for their customers, as they deposit their funds on these platforms. Exchanges usually charge fees for the use of their services.

From a money laundering point of view, the three risks associated with the use of exchanges are as follows:

—   They have not been obliged subjects in terms of money laundering prevention, with the entry of the new Royal Decree-Law 7/2021, on 27 April 2021, they are incorporated as new obliged subjects and a term of six months from the entry into force is established for their registration in the registry created.
—   The principals do not operate from Spain; however, Royal Decree-Law 7/2021 establishes that natural or legal persons, offering or providing services in Spain, must be registered in the registry.
—   They can be directly controlled by the money laundering organization.

Until now, as they are not obliged subjects, the customer identification policy has been defined by the company itself; therefore, the data required from customers has been very variable. Different international organizations, including the FATF, are recommending that new updates be made to international legislation on the prevention of money laundering to ensure that these companies can become obligated entities. Therefore, from now on, money laundering prevention controls should be established and certain standards in customer identification policies should be complied with.

### 5.3. Local Traders

These are people who advertise to exchange virtual currency for real currency, buying and selling cryptocurrency against cash as a co-trading exchange. The indications that would be linked to money laundering operations would be as follows:

—   Trading virtual currency against cash in significant volumes.
—   Conducting exchanges of virtual currency against cash using channels that involve paying high commissions and/or enduring worse exchange rates than alternative methods.
—   Exchanges made in unusual, anonymous locations. In many cases, sales and purchases are arranged through internet forums (p2p) and the exchanges take place physically to ensure that the buyer of the cryptocurrencies pays in cash and the seller simply provides his account password.
—   The purchase is made with direct profits from other illicit activities.

### 5.4. ATMS

These are ATMs that are used to introduce cash and transform it into the virtual currency that is sent to the user's wallet; it is also possible to sell cryptocurrency for cash. ATMs are usually installed in stores open to the public. In Spain, until 2021, ATMs have not been subject to licensing; therefore, their installation, maintenance and security measures are not regulated or supervised by any public body. The operation of a virtual currency ATM is as follows:

- The customer introduces cash into the ATM that he/she wants to transform into virtual money.
- The ATM sends the customer the equivalent amount in virtual currency, minus the commission charged for the operation, to the wallet indicated by the customer.
- The exchange linked to the cashier delivers the same amount of virtual currency to the operator at market price.
- The operator reloads the cashier's purse by transferring the virtual currency from the exchange.
- The operator collects cash from the ATM.
- Finally, the operator deposits the cash in the bank and makes a transfer to the exchange.

Until the effective entry into force of Royal Decree-Law 7/2021, the existence of Bitcoin ATMs increases the risk of money laundering, since they allow the direct introduction of cash at ATMs, without effective prior identification. Due to the entry into force of Royal Decree-Law 7/2021, which establishes that persons who provide services for the exchange of virtual currency into legal tender must be subject to preventive obligations, the installation of cryptocurrency ATMs, their maintenance, security measures, and the supervision they must carry will be regulated.

### 5.5. Online Videogames

The latest trend in money laundering is the use of online video games, which is very simple and appears in tutorials on different digital video platforms such as YouTube. There are numerous online video games that are used by criminals to launder their money. An example of this practice is the video game "Fortnite", which allows the purchase of its virtual currency, called V-Bucks, by acquiring cards containing a security code. These cards are resold on the Darkweb, in exchange for Bitcoins, to users of the video game at a price below the market price.

Another way of laundering money through this video game is using a system called Carding (illegitimate use of credit cards for profit to commit fraud), the procedure is very simple, it requires a previous illicit as it is the theft of cards to a third party, which are used to complete the profile of a powerful player who buys items in the video game with the stolen card and then sells them in exchange for virtual currency.

Once we have a cryptocurrency of illicit origin, there are different methods to launder it and introduce it into the traditional system, which are explained in this article.

### 5.6. Mixers

It can be translated as a mixer, it is a service offered by some virtual platforms to ensure that their clients can hide the origin of the cryptocurrencies that are registered in the blockchain, in fact, they are advertised as a way to have more anonymity in the operations. Such platforms mix the money of all the users they have and inter-exchange it, thus losing the traceability of transactions.

### 5.7. Suppliers of Services and Goods

There are more and more businesses that accept payment for goods and services directly with virtual currency, which entails a great risk, since it facilitates the laundering of money of illicit origin. A simple search on the internet shows websites where you can consult the list of businesses and companies that legally accept payments with virtual

currency, including those related to travel, leisure, free time, video games, digital stores, gift cards, computers, electronics, etc.

The mentioned businesses may make people think that cryptocurrencies are only used to buy goods and services with a limited, not very high, price. However, this statement is not true, since we can buy goods with a high market value, such as a house or a vehicle. It is fundamental that both parts, buyer, and seller, are convinced in carrying out the transaction in these conditions. Therefore, the value of the good must be fixed in advance; however, the volatility of virtual money means that, at the time of signing the sale-purchase, the price of the good can vary drastically in a matter of seconds, with both parties having to agree.

We can see how cryptocurrencies, especially Bitcoin, are increasingly accepted as a form of payment; however, the volatility of the currency and the possible cyber-attacks make the customer assume some risks when using it.

*5.8. Bitcoin Cards*

Another form of money laundering is carried out through the use of credit cards preloaded with balances in virtual currency, called "bitcoin to plastic" cards. These cards are loaded with virtual currency and are used to pay in euros or dollars. The essential characteristics of cryptocards are as follows:

— It is a prepaid Visa or Mastercard, issued by a card issuer that is licensed to operate.
— It is a card in euros, dollars, or other official currency.
— It is marketed to end users by a card provider, which has some kind of agreement with a card issuer or is a card issuer.
— There is a web page for the user, offered by the card provider.

These Bitcoin cards can be classified as either:

— Physical or virtual, depending on whether the card is issued physically or only virtually.
— Verified or unverified: It depends on whether the identity has been verified by the issuer of the card provider. When such verification has not been carried out or has not been efficiently checked, it is unverified.

These credit cards, preloaded with balances in virtual currency, are used for online payments or vendors that accept payment with Visa or Mastercard. They are also used for cash withdrawals through ATMs. Both methods are the almost perfect way to launder money, as they leave virtually no trace. Depending on the level chosen, it is not necessary to provide any documentation, or, if it is necessary to provide it, false documentation, or that of a third party can be sent. This makes it difficult to analyze the transaction, making it impossible to know the real owner of the card, and favoring the opacity of the operations carried out.

Table 1 shows a summary of the methodologies we have seen above.

**Table 1.** Summary of money laundering methodologies.

| |
|---|
| 1. Cryptocurrency mining |
| 2. Exchanges |
| 3. Local traders |
| 4. ATM |
| 5. Online videogames |
| 6. Mixers |
| 7. Suppliers of services and goods |
| 8. Bitcoin cards |

Source: own elaboration.

To conclude the section on methodologies, it should be noted that, in many of the types of money laundering through the cryptocurrencies analyzed, front men, shell companies, or tax havens can be used to hinder a possible investigation, making the transactions even more anonymous and opaque.

For example, a criminal organization wants to launder money of illicit origin. To do this, it buys a property with cryptocurrencies, through an instrumental company located in Spain, of which, in turn, 100% of the shares are owned by a company in Panama (tax haven) and the partner of this Panamanian company is a straw man. This simple scheme can be complicated as much as you want, putting different companies in between from different countries. When the investigators request information from a company to a tax haven, they will not provide it or it will be incomplete, thus hiding the real ownership of the property and if they obtain it, they will have to request the information again to a second tax haven, etc., thus making it impossible to know the real owner of the property.

## 6. Conclusions

In this article, two different topics have been discussed, money laundering and cryptocurrencies. These areas have, as a coinciding element, the specialization of criminal organizations that, in view of the new regulations for the prevention of money laundering, need new methods to give the appearance of legality to these funds of illicit origin.

In the traditional banking system, there is an intermediary, the bank, and a supervisory authority, whereas in cryptocurrencies these actors disappear, there is no central regulatory authority, although there are opinions in both directions: the ECB ensures that anonymity can be completely maintained and others claim that, since an identifier is required to perform transactions and this is associated with the wallets, this implies not having complete anonymity (Navarro Cardoso 2019). It is true that identifiers are not personal data, which facilitates functioning as a community, with the users themselves validating decisions through a blockchain system. "It is extremely difficult to achieve the nominal attribution of a bitcoin address or wallet, unless there have been security failures in the operations carried out" (Navarro Cardoso 2019, p. 13). Technological development is fundamental in the emergence of cryptocurrencies, surfacing to avoid dependence on traditional financial systems. As cryptocurrency technology continues to evolve, so will criminals, developing and improving new techniques that allow them to launder money in total opacity, without being able to trace the traceability of the virtual currency.

Currently, cryptocurrencies are being used by some criminal organizations, either directly through the organization's own technical staff, or by contacting criminal groups dedicated exclusively to laundering the money of other organizations. Thus, the US Department of Justice (2018, p. 123) states, "In recent years, money laundering through virtual currencies, such as Bitcoin, has become more prominent, as it allows Transnational Criminal Organizations to transfer illicit proceeds internationally and with more security than traditional cash transactions".

The characteristics of cryptocurrencies, especially the elimination of intermediaries and the privacy it provides, make it a very attractive and novel product for these organizations, using it as a means of payment in some illicit activities and as a method to launder illicitly obtained profits. The payment of goods and services through cryptocurrencies is becoming more and more accepted, as well as ATMs that allow the exchange of trust money for Bitcoins and vice versa, facilitating in both cases, for criminal organizations, the integration of "dirty money" into the legal circuit. Likewise, if there is an agreement between two people, a direct exchange of cryptocurrencies can be carried out, as well as the purchase and sale of houses and vehicles.

One of the options available to criminal organizations is to invest in the purchase of expensive computer equipment and set up a mining plant, providing cryptocurrencies, which can later be transformed into goods and services.

Until the publication of Royal Decree-Law 7/2021, there was no regulation on virtual currencies in Spain. Therefore, it was possible to send and receive cryptocurrencies, almost

with total anonymity, and there were no obliged subjects, nor a preventive or repressive regulatory framework. It is important to be aware of the fifth money laundering directive, as well as the FATF guide published in 2019, on Virtual Assets. Normally, FATF publications end up being translated at the European level, in a directive, which is finally transposed into our legal system.

In order to keep up to date and have a prospective vision of the changes that may occur in our financial system, it is essential to consult and study in depth the database provided by the FATF, as an updated, multidisciplinary, and almost inexhaustible source of resources.

The FATF guide, the fifth directive, and Royal Decree-Law 7/2021 define virtual assets and virtual asset service providers, establishing the obligated parties and a regulatory and supervisory body. They are based on a risk approach, in which the regulated entities must adopt customer due diligence measures. A preventive and repressive system must be established, expressly mentioning international cooperation as a means of information exchange. Royal Decree-Law 7/2021 establishes a Registry of service providers for the exchange of virtual currency for fiat currency and the custody of electronic wallets, constituted at the Bank of Spain, which specifies who must be registered.

One of the fundamental characteristics of cryptocurrencies is the privacy of the users or pseudo-anonymity. When all the measures mentioned in the Royal Decree-Law 7/2021 are implemented, it will be appreciated if the established regulation is accepted by the community or if there is some variant that tries to leave aside the approved regulations.

We also hope to convey that criminal organizations are often one step ahead of the legal system and the authorities that investigate them, and that it is necessary to change the rules of the game with strong legislation on the prevention and repression of the world of virtual currencies. It is essential that the investigating authorities have the necessary legal tools and resources to fight this new form of crime.

Continuous training in this area at the level of law enforcement, prosecution, and judicial system, as well as the creation of specialized prosecutors and judges who directly instruct the investigations or advise their colleagues as collegiate bodies, is essential to start fighting against the new crime that we have and that awaits us in the XXI century.

To conclude, we would like to point out that it would be a mistake to consider that most cryptocurrency transactions have an illicit origin or that the authorities should prohibit the use or development of these digital assets. In fact, this type of illicit use of cryptocurrencies is probably a minority. In this regard, it should be noted that cryptocurrencies are the newest monetary phenomenon of the 21st century and their development is bringing enormous benefits to our economies, as follows: new possibilities of exchange between companies and people all over the world with very low transaction costs, the new way of protecting one's own assets in situations of financial repression in various countries such as Venezuela or Argentina, the healthy competitive pressure introduced on the monetary authorities to ensure the purchasing power of state currencies and not to fall into the temptation of irresponsible monetary policies, the impetus and inspiration given to States to modernize fiat money through the Central Bank Digital Currencies (CBDC) and also, as a derivative consequence, the invention of blockchain technology and its innumerable applications in almost all sectors. For this reason, in our opinion, legislators and law enforcement agencies in different countries must use all possible means to prevent and prosecute money laundering through the use of cryptocurrencies, while taking care not to interfere in the development of this new technology.

**Author Contributions:** Conceptualization, C.d.R.; Formal analysis, D.S.-B., C.d.R., S.L.N.A. and M.Á.E.F.; Investigation, D.S.-B.; Supervision, S.L.N.A. and M.Á.E.F.; Validation, S.L.N.A.; Writing—original draft, C.d.R. All authors have read and agreed to the published version of the manuscript.

**Funding:** The APC was funded by the incentive granted to Sergio Luis Náñez Alonso by "Laws" journal of MDPI group.

**Institutional Review Board Statement:** Not applicable.

**Informed Consent Statement:** Not applicable.

**Acknowledgments:** The authors would first like to thank the Laws journal of the MDPI group for the incentive granted to support the APC. Second, the authors would like to thank the Catholic University of Avila for the monetary incentive derived from this publication.

**Conflicts of Interest:** The authors declare no conflict of interest.

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
