# Peer review of "Cryptocurrencies and Fraudulent Transactions: Risks, Practices, and Legislation for Their Prevention in Europe and Spain"

_laws_

Round 1
Reviewer 1 Report
This is a well-written manuscript. It’s also very informative and well structured. But some parts of the manuscript have to be reconsidered.
More specifically:
- From line 1 to line 107 the bibliography is set to emphasize Spanish literature on the topic of cryptocurrencies. Literature analysis is too domestic. There are no internationally relevant bibliographical references. I suggest that the bibliography be properly integrated in order to remedy this lack.
- I suggest that already in the introduction it is necessary to insert a short definition of cryptocurrency.
- lines 24-25: apodictic sentence, the author should briefly describe the positive aspects of cryptocurrencies.
- lines 85-95: the bibliographic reference on the origins of the cryptocurrency is insufficient. I recommend an integration.
- line 121: there is a misprint. The same word is repeated twofold.
- chapter 3: EU legal framework is insufficient. Although de jure condendo, the Pilot project by the European Commission is not mentioned.
- chapter 4 is too long. It’s too merely descriptive. I suggest of shorter it.
- paragraph 5.8, lines 650-682, should be shortened.
Author Response
Dear reviewer, good morning. We, the authors, would like to thank you for your time, as well as for the valuable advice you have given us. Now the manuscript is much more complete thanks to you. To facilitate the changes made, we have marked them in yellow.
From line 1 to line 107 the bibliography is set to emphasize Spanish literature on the topic of cryptocurrencies. Literature analysis is too domestic. There are no internationally relevant bibliographical references. I suggest that the bibliography be properly integrated in order to remedy this lack.
- Thank you for your suggestion. As you can see, throughout the introduction section, we have inserted new references following your advice. Subsequently, they also appear in the bibliography. We have also included mention of the future EU regulation called MiCA but in section 4.
I suggest that already in the introduction it is necessary to insert a short definition of cryptocurrency.
- Thank you for your suggestion. As you can see, we have introduced a definition of cryptocurrency. First paragraph of the introduction.
lines 24-25: apodictic sentence, the author should briefly describe the positive aspects of cryptocurrencies.
- Thank you for your suggestion. As you can see, the lines indicated have been revised and reordered to have a more appropriate meaning.
lines 85-95: the bibliographic reference on the origins of the cryptocurrency is insufficient. I recommend an integration.
- Thank you for your suggestion. As you can see, following your recommendation the indicated lines have been integrated.
line 121: there is a misprint. The same word is repeated twofold.
- Thank you for your suggestion. As you can see, following your recommendation this situation has been corrected.
chapter 3: EU legal framework is insufficient. Although de jure condendo, the Pilot project by the European Commission is not mentioned.
- Thank you for your suggestion. As you can see, following your recommendation this situation has been corrected. Reference to MiCA has been included in section 3, end.
chapter 4 is too long. It’s too merely descriptive. I suggest of shorter it.
- Thank you for your suggestion. As you can see, following your recommendation, paragraph 4 has been redrafted, reducing part of the text that has been crossed out.
paragraph 5.8, lines 650-682, should be shortened.
- Thank you for your suggestion. As you can see, following your recommendation, the lines indicated have been revised and rewritten.
Again, the authors thank you for your advice. We hope that the changes are to your liking and that we can continue with the publication process.
Best regards,
The authors
Reviewer 2 Report
- The structure of sentences needs to be corrected throughout the document, including punctuations. The citation in parentheses in the middle of the sentences needs to be deleted throughout the document. Otherwise, the sentences appear fragmented and incomplete. For example, “In the study by (Farrugia et 28 al., 2020), they analyzed in the Ethereum network a dataset of 4,681 accounts of which 2,179 29 were illicit; or in the case of (Hornuf et. Al. 2021).” This is just one example, and the authors need to fix these issues throughout the document.
- Also, the reference to et. Al. should be et al., 2021. These have to be corrected in the document.
- The use of acronyms such as ICO, DAO, CBCD, FATF, and FBI needs to be spelled out the first time when they are used.
- When referencing the order of items, the author used first, and secondly- It should be First and second, etc.
- Direct quotes require page numbers.
- The authors stated, “According to a ruling of the Supreme Court (2014), money laundering can be defined...Is the statement referring to the Spanish Supreme Court?
- Also, there are several short (one or two sentences)
- Citations do not conform to any particular citation format
Author Response
Dear reviewer, good morning. We, the authors, would like to thank you for your time, as well as for the valuable advice you have given us. Now the manuscript is much more complete thanks to you. To facilitate the changes made, we have marked them in yellow.
The structure of sentences needs to be corrected throughout the document, including punctuations. The citation in parentheses in the middle of the sentences needs to be deleted throughout the document. Otherwise, the sentences appear fragmented and incomplete. For example, “In the study by (Farrugia et 28 al., 2020), they analyzed in the Ethereum network a dataset of 4,681 accounts of which 2,179 29 were illicit; or in the case of (Hornuf et. Al. 2021).” This is just one example, and the authors need to fix these issues throughout the document.
- Dear reviewer, following your advice, changes have been made throughout the manuscript regarding citations, parentheses and citation. They are indicated in yellow throughout the text.
Also, the reference to et. Al. should be et al., 2021. These have to be corrected in the document.
- Dear reviewer, following your advice, this change has been made where necessary, throughout the document
The use of acronyms such as ICO, DAO, CBCD, FATF, and FBI needs to be spelled out the first time when they are used.
- Dear reviewer, following your advice, this change has been made where necessary, throughout the document
When referencing the order of items, the author used first, and secondly- It should be First and second, etc.
- Dear reviewer, following your advice, this change has been made where necessary, throughout the document. Lists have been prepared, following the order first, second...
Direct quotes require page numbers.
- reviewer, following your advice, this change has been made where necessary, throughout the document.
The authors stated, “According to a ruling of the Supreme Court (2014), money laundering can be defined...Is the statement referring to the Spanish Supreme Court?
- Dear reviewer, following your advice this change has been made. Mention has been made that this is the Spanish supreme court.
Also, there are several short (one or two sentences)
- Dear reviewer, following your advice, this change has been made when possible, because in some cases it has not been possible, given the summary nature of the measure made by the authors, as for example when we talk about the future European directive MiCA.
Citations do not conform to any particular citation format
- Dear reviewer, following your advice this change has been made. The bibliography has been entirely revised to conform to the format of the journal when it did not comply.
Again, the authors thank you for your advice. We hope that the changes are to your liking and that we can continue with the publication process.
Best regards,
The authors.